Trichocera maculipennis (Diptera)—an invasive species in Maritime Antarctica

Potocka Marta mpotocka@ibb.waw.pl 1
Krzemińska Ewa 2
1 Department of Antarctic Biology, Institute of Biochemistry and Biophysics Polish Academy of Sciences , Warsaw , Poland
2 Institute of Systematics and Evolution of Animals, Polish Academy of Sciences , Kraków , Poland
Ferrenberg Scott
Electronic publication date: 2018 Aug 14
Publication date: 2018
Volume: 6
Electronic Location ID: e5408
Received 2018 Feb 22; Accepted 2018 Jul 18
Copyright: ©2018 Potocka and Krzemińska
Copyright year: 2018
Copyright holder: Potocka and Krzemińska
License: This is an open access article distributed under the terms of the Creative Commons Attribution License, which permits unrestricted use, distribution, reproduction and adaptation in any medium and for any purpose provided that it is properly attributed. For attribution, the original author(s), title, publication source (PeerJ) and either DOI or URL of the article must be cited.
License URL: https://creativecommons.org/licenses/by/4.0/

Keywords: King George Island, Antarctica, Trichocera, Non-native, Invasive species, Biodiversity, Insects, Alien species

Funding: The authors received no funding for this work.

==============================
Antarctica, with its severe conditions, is poor in terrestrial fauna species. However, an increase in human presence together with climate change may cause an influx of non-native species. Here we report a significant increase in colonized area of one of the few known invasive species to date in Antarctica. Non-native flies of Trichocera maculipennis have been recently observed in the Admiralty Bay area on King George Island, South Shetlands Islands, West Antarctica, 10 years after its first record in Maritime Antarctica (Maxwell Bay, King George Island). Its rapid spread across the island, despite geographic barriers such as glaciers, indicates successful adaptation to local environmental conditions and suggests this species is invasive. The mode of life of T. maculipennis, observed in natural and anthropogenous habitat and in laboratory conditions, is reported. The following adaptations enabled its invasion and existence within the sewage system in Antarctic scientific stations: the ability to survive in complete darkness, male ability to mate on the substrate surface without prior swarming in flight, and adaptation of terrestrial larvae to survive in semi-liquid food. Possible routes of introduction to Antarctica and between two bays on King George Island are discussed, as well as further research leading to the containment and eradication of this species.

Introduction

Antarctica has been isolated since the Early Miocene, and most of its native organisms evolved in harsh polar conditions without contact with other ecosystems (Murphy et al., 2013). Terrestrial ecosystems in Antarctica are limited to ice-free areas, consisting of only 0.2% of the whole continent’s surface, such as nunataks and seasonally snow-free regions (Convey, 2007; Hughes & Pertierra, 2016; Burton-Johnson et al., 2016). These areas are found mostly along the coast, especially the Antarctic Peninsula and surrounding archipelagos, and at several oases in East Antarctica (Levy, 2013). Low temperature, strong wind, short vegetation season and limited ice-free areas make Antarctica inhospitable for most terrestrial organisms. Therefore, biodiversity of ice-free regions is very low, with flora represented by only two native higher plants, and fauna limited mostly to micro-invertebrates and two species of macro-arthropods, Belgica antarctica Jacobs, 1900 and Parachlus steinenii Gercke, 1889 (Smith, 1984; Frenot et al., 2005; Hughes & Pertierra, 2016). The ecological structure of communities is simple, with limited competition for resources (Hughes et al., 2013).

Due to increased interest in Antarctica, which began during the early 20th Century, and the growth of scientific and tourist activity in Maritime Antarctica in the last 20 years (Lee, 2008), together with climate change, the region has become vulnerable to non-native species introduced accidentally. Increasingly, non-native species have become a major threat for the Antarctic communities (Hughes et al., 2015). Most alien species cannot survive in Antarctic conditions, but several have been able to adapt to new territories (Hughes, Greenslade & Convey, 2017; Olech & Chwedorzewska, 2011). Other species may remain in the vicinity of human settlements, where they can reproduce in more favourable conditions (Hughes et al., 2005). In 2013, Volonterio et al. (2013) observed abundant Diptera in the sewage system of the Uruguayan Artigas Station, in the Fildes Peninsula on King George Island. The flies were identified as Trichocera maculipennis Meigen, 1818, known from the Northern Hemisphere. Attempts to exterminate were unlikely to be successful, most likely because this species was already established in the natural environment. This was indicated by observations of adult specimens flying outside the buildings, so presumably they were able to recolonise the sewage facility at the Artigas Station (Volonterio et al., 2013).

In October 2017, at the Polish Antarctic Arctowski Station in the Admiralty Bay, King George Island, non-native Diptera, larvae and adults, here identified as T. maculipennis, were found in the sewage system. This suggests that the population of flies observed at the Maxwell Bay area in 2007 (Volonterio et al., 2013; Korea Uruguay Chile United Kingdom, 2017) had dispersed to the next bay more than 20 km away over a glacier barrier (Fig. 1). This is the first record of dispersal of flying, non-native insects introduced by human activity to Antarctica.

Figure 1 Map of King George Island with scientific stations affected by T. maculipennis (M Potocki).

(1) Artigas Uruguayan Station, colonized in 2006; (2) Frei Chilean Station, colonized in 2009; (3) King Sejong Korean Station, colonized in 2013; (4) Arctowski Polish Station, colonized in 2017.

The aim of this study is to investigate the continued expansion of T. maculipennis to different ice-free areas near the Polish Antarctic Arctowski Station on King George Island, so that the risk of further dispersal can be estimated, and appropriate management action can be implemented. As background, the life cycle and habits of this species in a synanthropic habitat and in laboratory conditions are compared.

Material and Methods

The presence of possible same species of alien flies was noticed for the first time on the Polish Antarctic Arctowski Station, King George Island, South Shetland Islands, on October 23rd 2017. Routine inspections in previous years (2015/2016) had not revealed their presence (E Przepiórka, 2016, unpublished data). The insects, in large numbers, were found in one holding tank of the sewage system near the entrance to the main building of the station (Fig. 2). The sewage system at the Polish Antarctic Station consists of four independent systems with pipe heaters. Three other more distant tanks were free of insects. Thirty adults and 20 larvae were collected at random and preserved in 96% ethanol.

Figure 2 Polish Antarctic Arctowski Station on King George Island, Admiralty Bay and its sewage systems colonized by non-native flies (M Potocki).

For identification purposes, specimens of T. maculipennis from various European localities were compared (listed in Volonterio et al., 2013). Only the taxonomic description, based on morphological data, was possible, as there is no molecular data for this species (GenBank).

Results

Systematics and identification

The diagnostic characters of adult flies, males and females, were given in Volonterio et al. (2013). Their venation and pattern of spots on wings (Figs. 3A, 3B) were used for species identification. Since the larvae in the sewage system appeared abundant, and are the potential target of eradication, their morphology and mode of life are summarised herein.

Figure 3 Identification of T. maculipennis (photo: E Krzemińska).

(A) wing of the specimen from the Polish Antarctic Arctowski Station, Admiralty Bay; (B) for a comparison, a wing of the specimen from terra typica (Austria, cave; the middle part seems somewhat misshaped because the wing is dry). Larva: arrangement of segments (C); a characteristic, elongated and strongly sclerotized head capsule; anterior spiracle (D); scheme of anal end with four anal lobes surrounding two large spiracles, in posterior (E) and lateral (F) view. Abbreviations: dl, dorsal lobe; s, spiracle; vl, ventral lobe.

The larva of T. maculipennis (Rhynehart, 1925; Brindle, 1962) is similar to those in all congeneric species (Figs. 3C–3F). In the genus Trichocera the larva has a strongly sclerotized, dark head capsule, partially retractable into the first (prothoracic) segment. The first four segments behind the head are secondarily divided in two; the next six into three parts. The secondary segmentation is present even in the first instar, although weakly expressed. The last, 11th segment, bears four fleshy lobes, two dorsal and two larger, ventral ones; between them, a pair of large spiracles is situated. Although trichocerid larvae are generally known to be terrestrial, the larvae of this species may thrive in a semi-liquid or liquid substrate because they can raise the anal end with spiracles above the surface; the four anal lobes (Figs. 3E, 3F) are able to adhere to the spiracles to cover them when necessary (Karandikar, 1931).

Distribution

Trichocera maculipennis is widely distributed over the Holarctis. Unlike most congeneric species, it seems to be tolerant to both warm and cold climates. The range of this species in the Northern hemisphere is wide, from the Arctic to the most southern regions of the Mediterranean area (Azores, Canary Islands; Dahl, Krzemińska & Baez, 2002), and in the Far East, to the Kuril Islands (Petrašiūnas & Paramonov, 2014). All records from most northern, Arctic localities, confirm the synanthropic character of the species in these harsh conditions (Iceland, Bear Island, Greenland and Jan Mayen; Dahl, 1957; Dahl & Krzemińska, 2015). In the Southern Hemisphere, T. maculipennis is known from two locations: the Kerguelen Islands (Séguy, 1940; Séguy, 1953) and King George Island (Volonterio et al., 2013). The species was most likely introduced to the Kerguelen Islands by human agency, and dispersed from one of the ports (Dahl, 1970b).

Notably, invasions of two other Holarctic trichocerids to the Southern Hemisphere resulted in established populations which have existed there for c.100 years: Trichocera regelationis L. in South Georgia, first recorded by Bréthes (1925) and Dahl (1970a), and T. annulata Meigen in Australia (Alexander, 1926) and New Zealand (Edwards, 1928). These early introductions were accomplished most likely with soil taken as a ship’s ballast. Both species are closely related with T. maculipennis (Krzemińska, 1999), as discussed in Volonterio et al. (2013).

Observations on the life mode of T. maculipennis at the Polish Antarctic Arctowski Station

The dipterans, discovered in October 2017 in large numbers, were active, flying inside the sewage system. The temperature under the lid of the sewage tank fluctuated from +1 °C to +3 °C, while the outside temperature was −3.5 °C to −2 °C. Only a few adult individuals were noticed outside the facility, at a distance of less than 50 cm. They most likely escaped accidentally when the lid was open. No larvae or pupae were observed outside, whilst adults stayed on the ground or snow, and did not fly (Fig. 4E).

Figure 4 T. maculipennis from the Polish Antarctic Arctowski Station, Admiralty Bay (photo: E Krzemińska (A–D, G) and E Przepiórka (E–F)).

Various stages of female maturity: young female with eggs not fully developed yet (A); a mature female filled with ripe eggs (B); and old female devoid of eggs (C). (D) striped abdomen, posterior margins of segments are lighter. (E) a female on snow, in vicinity of the sewage. (F) swarming on the substrate surface. (G) larvae: on the left, a possibly third instar, remaining larvae are fourth instars in various sizes; note the dark brown color gradually developing together with age (size) from left to right.

The sample collected contained three females, each at different stages of abdominal maturity (Figs. 4A–4C). The mass presence of the males on the surface of the substrate (Fig. 4F) was unexpected, as males of other trichocerid species swarm in flight, singularly or in groups. The presence of specimens with striped abdomens (Fig. 4D) is notable; this feature appeared variable even in a very small sample (Figs. 4A–4C: A, C with stripes, B, uniformly brown). Similar variation was observed previously in the Uruguayan Artigas Station (Volonterio et al., 2013). This feature seems to occur only sporadically in T. maculipennis, as discussed in Volonterio et al. (2013), therefore this observation seems important when considering a probable source of this species at the Polish Station.

Larvae were not buried, staying on the wet wooden surface inside the sewage cover box. One larva was most likely third instar, whilst all others were fourth instars (Fig. 4G). The dark brown color of the largest and possibly oldest larvae is noteworthy. It most likely comes from a gradual, passive absorption of coprosterols (present in human excrements; the breakdown product of cholesterol) which make the larvae inconspicuous on the substrate of the same color. Nowhere in the literature has such a dark color been recorded in larvae of this or other species of Trichocera; the trichocerid larvae are usually whitish or grey in colour.

Biology

For the successful eradication of this species, knowledge on the biology of its life stages (egg, larva, adult) is essential; particularly their resistance or vulnerability to external conditions and the habits which enable this particular species to spread outside its genuine distribution.

All stages of Trichocera species are vulnerable to desiccation and freezing or warming; according to Dahl (1969), temperatures lower than −10 °C and higher than 32 °C kill adults and larvae. Steady temperature oscillating around 0 °C (−2 °C to +5 °C) seems especially important for development of the eggs and larvae of this species, as may be deduced from its preference of synanthropic life in habitats such as mines (even to 400 m deep and in complete darkness; Grimshaw, 1906), cellars, and storage houses, but also in natural caves where temperature conditions are more or less stable (Plachter, 1983; records since XIX century are reviewed by Petrašiūnas & Weber, 2013). These habitats also provide rich sources of food for the larvae: rotten vegetables, animal droppings (especially guano in the caves), compost and sewage waste. According to Karandikar (1931), the duration of life cycle strongly depends on the environmental conditions (Table 1); in the laboratory, the cycle may be completed in c. 40 days, while outside, in a synanthropic habitat, it may last up to one year. This discrepancy is difficult to explain. Extended appearance of adults in anthropogenous habitats can be a result of repeated, fast developing generations, or by gradual emergence of females of one generation.

Table 1 Duration of life stages of Trichocera maculipennis in the laboratory (days, temperature stable at −1 °C) and outside, in synanthropic habitat (distillery tanks; indicated are only the months when the specimens were noticed).

All specimens originated from the same population in Scotland. After Karandikar (1931).

Stage	In laboratory	Around distillery	
egg	5–6	?	
larva	25	III–IX	
pupa	7	?	
adult	14	IV–IX	
Duration of the cycle	c. 40 days	Several months to 1 year	

Other information on the life mode of T. maculipennis is restricted to casual observations of the adults’ appearance. In the Arctic region, the adults appeared in July to August (Dahl, 1967a); in the subarctic region in June to August, and in Sweden and Finland from early spring to September (Dahl, 1967b). In the Southern Hemisphere, in Kerguelen, the adults appear during the Antarctic summer (January to March), winter (July to August) and spring (September), which is nearly all year round. Some adults were found in caves, and others near human settlements (Port Jeanne d’Arc, Port aux Français).

Discussion

Antarctic Hexapoda are represented only by a few species of Collembola and two dipterans: B. antarctica and P. steinenii (Convey & Block, 1996). Every appearance of a new non-native species, especially one successfully reproducing in the area, should be considered as a serious threat for Antarctic terrestrial ecosystems and their biodiversity (Committee for Environmental Protection, 2016). Due to these concerns, one of the strictest pieces of legislation concerning non-native species was adopted: the Protocol on Environmental Protection to the Antarctic Treaty. The Protocol prohibits the intentional introduction of non-native species to Antarctica, except in accordance with a permit issued under prescribed circumstances, and stipulates their removal or disposal prior the expiration of the permit.

The case of non-native T. maculipennis found in Admiralty Bay on King George Island at the Polish Antarctic Arctowski Station, and 10 years earlier in the Maxwell Bay (Fildes and Barton Peninsula) on the same island (Fig. 1), was one of the first reports of successful reproduction of flying insects in Antarctica outside station buildings, but still within human structures where the climate differs greatly from that outside. In this section, we discuss two stages of this invasion: on to Antarctica, and a subsequent migration on the island. Countermeasures to be implemented against further spreading are proposed, as well as the eradication procedures applied so far.

Invasion of King George Island

Until now, in most cases, the presence of flying, potentially invasive, invertebrates in Antarctica corresponded with ships’ or aircrafts’ operations during supply periods for scientific stations (Hughes et al., 2011). The Chilean Frei Station’s airport on Fildes Peninsula, King George Island, may be considered as a potential transport hub for non-native species between South America and other regions of King George Island, other islands in the South Shetlands archipelago, and Antarctic Peninsula region. Other sources of spread must also be considered, such as ships with tourists or personnel of stations, and above all, the cargo of multiple operators from their home countries. Many studies (e.g., Lee & Chown, 2009; Tsujimoto & Imura, 2012; Chwedorzewska et al., 2013) suggest that almost all cargo for scientific stations in South Shetlands Islands can be a potential vector for non-native organisms.

The most likely way of introduction to Antarctica would be by the cargo of root vegetables stored in the ships going directly from Europe or Asia, as there is no evidence of this species in South America. Potatoes, swedes, and other tubers are known to be the food source for the larvae of T. maculipennis (Dahl & Krzemińska, 2015). However, females do not lay eggs in fresh bulbs or tubers, neither can any larvae be found within them because the tissue of such vegetables is too hard. The attack is possible only after vegetables have been softened by previous infection by molds and bacteria, therefore a significant threat is posed if the vegetables show traces of rotting. Deep freezing of vegetables would be lethal to trichocerids in all life stages, and would eliminate future introduction of non-native insects to Antarctica.

Migration between the stations and possibility of establishment in natural habitats

The Polish Antarctic Arctowski Station is the fourth station on King George Island invaded by T. maculipennis (Fig. 1). The flies were observed first in the Fildes Peninsula region at the Uruguayan Artigas Station in 2006, then in 2009/2010 at the Frei Chilean Station, and in 2013/2014 in the Korean King Sejong Station on the Barton Peninsula (Peter et al., 2013; Korea United Kingdom Chile Uruguay, 2016). In January 2015 adult specimens were caught also inside the Escudero Station (P Convey & T Contador, pers. comm., 2015). Separate introduction events to these stations seem less probable than spreading of one original population.

Methods of spread are currently unknown. The species can fly in temperatures around 0 °C, thus flight seems an obvious possibility. However, Trichoceridae are poor flyers, and flight cannot take place in windy conditions (Dahl, 1965). Nevertheless, a passive dispersal on wind currents should be considered. Observations of several other congeneric species suggest that the flies may spread by walking combined with short episodes of flight over the surface - either ice-free or snow covered surfaces (e.g., Hågvar & Krzemińska, 2008). A recommended method of checking dispersal ability would be to monitor areas between stations, combined with pitfall traps which are known to attract the trichocerid females.

Recolonisations of sewage systems at the Artigas Uruguayan Base in Maxwell Bay, after unsuccessful eradication attempts, indicate this species’ ability to survive in natural environments (Volonterio et al., 2013). A breeding place and substrate to lay eggs in, alternative to those offered by human activity, may be the guano of birds or pinnipeds, with ornithogenic soil providing a temperature higher than freezing point (Zdanowski, Zmuda & Zwolska, 2005). Some shelter may also be provided by the low vegetation, which includes mosses and vascular plants: native hair grass Deschampsia antarctica and pearlwort Colobanthus quitensis, and an invasive bluegrass, Poa annua L. (Olech & Chwedorzewska, 2011; eradication of the grass is now in progress; Galera et al., 2017).

Human traffic between stations and the transfer of eggs or larvae on shoes or personal equipment could be another alternative form of invasive species dispersal. During the season in 2016/2017, the Polish Antarctic Station was visited by 52 scientists from different countries, including Brazil, Chile and Argentina. Most of them travelled to King George Island from South America via Chilean Frei Station’s airport in Maxwell Bay. The station’s personnel would also visit other bases on occasion during holidays, many times taking fresh food as gifts, a potential source of larvae. There is no data showing the intensity of this kind of human traffic between scientific stations. The Arctowski Station was also visited by tourist ships coming from South American ports; in the 2016/2017 season, 649 people landed on the shore of Admiralty Bay (Annual Report Polish Antarctic Arctowski Station 41. Antarctic Expedition 2016/2017; B Matuszczak, 2016/2017, unpublished data). Tour operators in Antarctica have strict biosecurity measures including boot washing and cleaning personal equipment, and this way of introduction is least likely. Furthermore, the eggs of all trichocerids are very delicate, short–living and immediately desiccate outside wet substrate. However, some cruise ships send fresh food as gifts to the stations they visit, so that may be a risk of repeated introduction of alien species too.

On the South Shetland Islands, where environmental conditions are similar to those experienced by T. maculipennis in the Northern Hemisphere (Dahl, 1970a), the species has the physiological capacity to colonise suitable habitats in the Antarctic Peninsula region, as it had done before in the Kerguelen Islands where it was observed within and outside human settlements (Dahl, 1970b). However, Kerguelen Island is 13° further north (49°S, compared to 62°S at the Admiralty Bay). The vegetation on that island, although low, is present all year round, and climatic conditions, although harsh and windy, are less severe than those in Admiralty Bay (Convey et al., 2018). The temperature on Kerguelen Island is higher than on King George Island (+1 °C in August and +6° C in January on Kerguelen Island, and +0.8 °C in January and −6 °C in August on King George Island; Petlicki et al., 2017). Presence and further spread of this species on King George Island indicate its adaptation to colder temperatures.

A case of colonisation similar to that of T. maculipennis was observed previously by Hughes et al. (2013) of an alien chironomid Eretmoptera murphyi Schaeffer, introduced accidentally from South Georgia to Signy Island, during a soil transplantation study in the 1960s (Edwards & Greene, 1973; Block, Burn & Richard, 1984). This species became invasive and acquired a biomass exceeding more than twice that of the entire native microarthropod species at the location where E. murphyi occurs at the Signy Island, and is thought to have a significant impact on nutrient cycling in the environment (Hughes et al., 2013).

Eradication in human settlements

Presence of flies in one of four sewage systems at the Polish Antarctic Station (Fig. 2) indicates an early stage of colonisation in the Admiralty Bay region, and immediate management actions are necessary to stop further spread. Previous unsuccessful eradication attempts of T. maculipennis at Uruguayan and Korean Station on Fildes Peninsula (Volonterio et al., 2013; Korea United Kingdom Chile Uruguay, 2016) have shown that one-off action would not cause total extermination.

Protocols implemented against an invasive sciarid fly (Lycoriella sp.) at the Australian Casey Station also appeared insufficient (Hughes et al., 2005), whilst the same species was effectively eradicated at the Rothera Station. The difference between living conditions of populations in both stations were significant. At Rothera, the flies were inside the building (in the alcohol bond store), while at Casey Station the pupae were hidden in nooks and crannies in the sewage system, and their eradication was hindered, so much so that they appeared resistant to the chemicals used. Synanthropic species are difficult to eradicate (Hughes et al., 2005), especially in such delicate ecosystems as terrestrial Antarctica, where very strict norms of using insecticides are to be obeyed.

The first steps of a comprehensive eradication protocol were implemented at the end of the season in 2017/2018 at the Arctowski Station. The protocol included several different methods targeting different life stages of invasive flies. Firstly, the affected sewage system was cleaned and treated with the solution of hydrochloric acid. The treatment appeared effective; the sewages are now free of insects, but reappearance can be expected as it had happened at the Uruguayan and Korean stations. Monitoring will be continued and the protocol repeated if needed.

As a next step (season 2018/2019), the locations of insect occurrence will be fumigated with insecticides containing lambda-cyhalothrin. This substance is a synthetic pyrethroid which disrupts insects’ nervous system. Insecticides containing proposed chemicals belong to the long-acting substances (up to six to eight weeks), and are used for controlling adults, larvae and eggs of insects. At the Arctowski Station they will be used only inside the sewage system, without contact with natural environment. The substance breaks down into harmless compounds after six to eight weeks. UV lamps and electric flying insect killer lamps will be used inside the sewage’s cover box and in the vicinity of the sewage system to control flying adult specimens.

It is necessary also to control the natural environment: trichocerids on flight, especially just above the ground, and walking on snow (Fig. 4E), and to implement and check the pitfalls. Flying trichocerids are easily collected with a net; the pitfalls will not eliminate the species, but may confirm its presence in the field. As there is no molecular data for T. maculipennis, either from the Northern hemisphere or Maritime Antarctica, the barcoding studies may establish the origin of its population on King George Island. Molecular studies are necessary to reveal the sources of fly introduction and to determine whether it was a one-off incident or possibly the result of multiple occasions. Further research on the potential influence of T. maculipennis on native fauna and their possible reproduction outside the research bases must be considered.

Subsequent complex measures, taken jointly by all parties, should be implemented as soon as possible. First steps to establish an action plan to manage the non-native flies in King George Island were taken jointly by Korea Uruguay Chile United Kingdom (2017) (ATCM XL–WP 26). This action should be extended and supplemented.

National operators of scientific stations on King George Island are aware of the problem of non-native flies, and together, are planning further works, starting with systematic monitoring. A proposed extermination protocol will be shared between operators and discussed.

Conclusions

This paper presents a report of an increase in distribution of one of the few known invading species to date in Antarctica. Invasive species may have a fundamental new ecological role in the ecosystem they occupy. It looks unlikely that control or eradication can be successful, other than through the application of strict and effective biosecurity measures by the national operators using King George Island and the South Shetland Islands. As a result, a complex eradication plan is necessary, and proposed in this paper. The aforementioned methods are appropriate to apply in a sewer system or inside buildings, but if T. maculipennis inhabits a natural environment then the protocol will be useless, and total eradication will be extremely difficult, if not impossible. Until there is no evidence of settlement of flies in a natural environment (larvae or eggs found outside of research stations), the proposed protocol is recommended. We also recommended consideration of a ban on the import of fresh root vegetables to the Antarctic region, or at least consideration of importing clean and properly secured vegetables from known suppliers. There should also be a total ban of transfer of fresh fruits and vegetables from unknown sources, such as gifts from passenger ships or other stations.

Successful long-term reproduction and range extension of a flying insect in Antarctica proves that non-native species constitute an increasing threat for the biota of one of the most isolated regions on earth. It is still unknown how dispersal occurred between ice-free regions of the island—Maxwell Bay and Admiralty Bay—and whether they spread naturally despite the geographical barrier of a glacier (least likely), or with people (and cargo) travelling between stations, or were introduced separately at each affected station. To answer that question, molecular studies are planned. Additionally, a common strategy of monitoring has already started by the national operators on King George Island. Control and eradication must be developed for King George Island, with strict biosecurity procedures stopping the further spread of this species.

We would like to acknowledge the help of participants of XLI and XLII Polish Antarctic Expeditions of the Polish Antarctic Arctowski Station (especially Bartosz Matuszczak and Ewa Przepiorka) for collecting material, Dr Anna Kidawa for discussions and advice which have led to the development of this paper, and Mariusz Potocki for map preparation. The authors wish to thank Peter Convey and the anonymous reviewer for helpful comments, and Kate Allberry for English correction.

Additional Information and Declarations

Competing Interests

Author Contributions

Data Availability

The authors declare there are no competing interests.

Marta Potocka and Ewa Krzemińska conceived and designed the experiments, performed the experiments, analyzed the data, contributed reagents/materials/analysis tools, prepared figures and/or tables, authored or reviewed drafts of the paper, approved the final draft.

The following information was supplied regarding data availability:

The material (T. maculipennis specimens) is stored in the Institute of Biochemistry and Biophysics, Polish Academy of Sciences in Warsaw, Poland.

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
