# Peer review of "Trichocera maculipennis (Diptera)—an invasive species in Maritime Antarctica"

_PeerJ, doi:10.7717/peerj.5408_

## Round 0.1 · original submission · Major Revisions

At this time, your manuscript has been reviewed by two experts who offer numerous, important comments that will improve this submission. Key among the comments made by both reviewers is the need to revise the entire manuscript to correct grammatical issues and improve the overall structure. Both reviewers also make a note that prior research regarding the introduction and attempts to control T. maculipennis must become more prominent in the introduction section of this manuscript.

In response to comments of Reviewer 1, I agree that it is a necessity that any previously published content used in this manuscript be properly identified and referenced, and that permission for reprint secured as necessary in the case of figures 3 and 4. A detailed response on this topic, with explicit statements of what is original to this current manuscript and what is drawn from previous work, is necessary. Also, Reviewer 1 makes note of an invasive mycocelid fly that offers an example of previous introductions and efforts to control the spread of exotic species in this ecosystem. Incorporating this information would enhance the context of this current work and inform readers as to the scope of invasive pressures affecting these remote research stations.

Finally, Reviewer 1 also notes the potential advances that could come from DNA based work with this organism. Determining source populations or potential genetic exchange among habitats would indeed be a substantial advance. Recognizing this potential in the discussion of the manuscript at a minimum would be helpful.

Reviewer 1 ·

Basic reporting

Review of “Trichocera maculipennis (Diptera) – a potential invasive species in maritime Antarctica” for PeerJ.
By Potocka and Krzeminska
This mss, which is long at nearly 450 lines with four plates of figures, documents a new location for an exotic fly on King George Island. The supposedly new information shows that the species can spread to scientific stations across glacial barriers. The manuscript also includes several pages on the systematics, life history and distribution of the species which has already been published by the second author with other authors (2013). Moreover, earlier documents, including a publically available Working Paper (CEP 10a, 2016) have already documented the new locations and its spread is not surprising as it has already been recorded outside station buildings in the natural environment. The inclusion of figures 3 and 4 are also unnecessary as all of them have been published before by Krzeminska. I do not consider therefore that the publication of this paper is justified in its current form as it only records a new locality for the fly and does not advance scientific knowledge of the species.
What I would have liked to have seen in this mss was a discussion of the attempts at eradication including details of methods used, why they failed, how the method could be improved by paying more attention to the vulnerability or not of different stages in the species’ life history. For instance,develop an eradication plan using not one method, but several different methods could be used, each targeting a different life stage. Information from attempts to eradicate a different species from the Australian Casey Station is relevant but is not mentioned. To make space for the eradication discussion I have marked on the text which sections should be deleted.
My detailed comments are below.
• The English is poor and clearly has not been checked by a native English speaker. Practically every manuscript I am sent to edit is authored by non-native English speakers who have not bothered to have their writing edited. This wastes a deal of my and other referees and editors’ times and is quite annoying. This mss is one such and although I have taken time to improve the text, there is more to be done. I urge the authors to elicit the help of someone or several people who know and have experience in writing scientific papers in well crafted English before submitting again.
• There are numerous omissions in particular any references to the exotic mycocelid fly in the sewage treatment plant (tanks and pipes) at Casey Station and authors seem ignorant of it (line 318). This fly was eventually eliminated I beleive, although initial attempts by engineers failed while an entomologist was consulted but their recommendations not followed. The problem at Casey was that the pupa was resistant to heat, cold, moisture, chemicals etc etc and hid in nooks and crannies in the sewage system. The authors do not mention this possibility with T. maculipennis pupae nor do they suggest that extermination is necessary and discuss methods (see above). Admittedly they recommend an action plan be prepared jointly by four countries but they do not suggest any method of removal nor any plan as to how these four nations will be able to produce one. I can see this being a ten year project meanwhile the fly expands its range. Control and extermination should start now.
• The authors seem convinced that the fly has colonised the additional stations by long term natural dispersal but I suggest this is unlikely. Despite the fly being apparently immobile in strong winds, there is no reason to suppose it could not have been dispersed on wind currents accidentally. The authors seem not to have considered this possibility but must do so and include in the paper the routes across and along the coast of the wind at different times of year.
• What is more likely by looking at the current records being only at scientific stations, is that it was human agency that dispersed the propagules. The authors do not document whether there was any human traffic between Stations but I suspect that there is and was. What about tourists for instance and “adventurers”. It could well be difficult to get this data but it must be added. Again this route is not considered by the authors .
• In the title, I do not know why the authors consider the species a potential invasive. Surely the new dispersal records show that it is invasive. Has it been documented as invasive in other parts of the world?
• Why were specimens not bar coded? Specimens from Kerguelen should also be barcoded as well as from each station on King George Island. I would be surprised if there are not barcodes from European specimens in GenBank for comparison. The origin of each population could then possibly be established.
• It is well known that fresh fruit and vegetables imported into the Australian Antarctic and subantarctic Stations are the source of introductions of exotic invertebrates. Indeed, some stations have banned the importation of these commodities for this reason. The mytocelid fly at Casey is a pest of mushrooms for instance. So the best method of preventing reinfestation, if it is eliminated, is not to import fresh root vegetables in particular. That will be cheaper in the long run than a succession of eradication efforts.
• The life history of the fly is described in detail much of which is not relevant to the situation on King George Island. Instead, information is needed on the conditions in the sewage plant, i.e. temperature and the tolerance of the species to polluted waters and especially if the pupa is hardy and cryptic (see above). Some information only is given of its behaviour in non Arctic locations.
• n dashes and not hyphens should be used for all spans i.e. page numbers. All references are given in the wrong format.
• The figures 1 and 2 are useful and of a sufficiently good standard to be included including a complete rewrite is needed with attention to all my comments above.

Experimental design

See above. n.a.

Validity of the findings

No findings - see above

Additional comments

See above. Total rewrite necessary and collection of more data.

Annotated reviews are not available for download in order to protect the identity of reviewers who chose to remain anonymous.

·

Basic reporting

The paper is clearly structured and appropriately referenced. My main concern is with the quality of the English language used, and this goes beyond suggesting a few sentences for specific editorial attention - basically the entire text needs to be very carefully edited by a first language English speaker, preferably with an ecological science background.

Experimental design

This is a largeyl descriptive study, so comments on experimental design are not appropriate. The subject of the paper is clear and well defined, and the importance both of the general research area (biological invasions in Antarctica) and of the specific example being reported are clear. The data being reported are appropriately described.

Validity of the findings

As an observational study, statistical analyses are not included or necessary, but the results are robust and clear. Some comments are included below about what I perceive as an area to include the 'balance' of the conclusion of the importance of sewage tanks in the dispersal/distribution of this species on King George Island as distinct from its possible/probable presence in the natural environment, from which it then opportunistically colonises these sewage tanks.

Additional comments

This is a very important report of a significant increase in distribution of one of the few known invading species to date in Antarctica, and of a species that may have a fundamentally new ecological role in the ecosystems it occupies, as well as one that looks extremely unlikely to have any possibility of control or eradication, other than through the application of strict and effective biosecurity measures by the national operators using King George Island and the South Shetland Islands. I would even suggest a sentence such as the previous would be appropriate to include in both the Abstract and as a conclusion of the paper. Also, although not part of the reviewing process, I would strongly urge the authors to report this new finding as an Antarctic Treaty paper - probably too late to do so at the imminent and abbreviated Buenos Aires meetings, so perhaps for the 2019 meeting in the Czech republic.

Introductory text: one general point that does not come through clearly to me at this stage in the ms is that in the original report of this species (Volonterio et al 2013) the suggestion was made explicitly that the species was actually established in the natural environment on KGI and from there recolonised the sewage facility at Artigas Station a couple of years after the first eradication. This is mentioned quite late on in the Discussion, but I think the earlier study should be credited with this earlier in the paper (as early as mention of this study at l68-71). Realising this also changes the balance between the new colonisation event being possibly related to direct human movement (eg through eggs on boots contained in rotting debris) and natural dispersal (eg a wind blown gravid individual - the answer will never be known, but I don't think the balance is quite right in the current text. It would perhaps also be appropriate to mention that, as far as I am aware, none of the operators (national operators, I believe tour operators mostly apply boot washing around landings) on KGI apply any biosecurity measures around personnel and cargo movement at present, despite several years' knowledge of this species' presence on the island. I am also not aware of any detailed surveys of occurrence/distribution in the wider ice free areas accessible from the airport at Frei, other than those carried out in the vicinity of Artigas Station a few years ago.

l48 - the most recent estimate of this (Burton-Johnston et al 2016) reduces this estimate to c. 0.2%

l160 - such discrepancies are not unusual, particularly in studies of invertebrates that may have a diapause stage in the natural environment

l186 - the form of any attempt to survey beyond the station ('outside') is not actually described?

Line 308: The eradication of Lycoriella was successful at Rothera, but not at Casey

Line 271-273: The first half of the sentence could be clearer (it wasn’t the biomass of the whole island, rather in the specific location where Eretmoptera occurs!)

l289 - the airport is clearly such a risk, but should also be noted that most cargo (including fresh food) arrives for the many operators on KGI and in the S Shetlands by ship, and again the transit time from southern S America is a small number of days, so provides another high risk area of operation.

l293 - the comment about not being resistant to higher temperature contradicts that a few lines above about the species being distributed in Europe as far south as the Mediterranean.

Peter Convey
British Antarctic Survey

---

## Round 0.2 · Major Revisions

Both of the expert reviewers find this version of your manuscript to be much improved and hope to see continued efforts to further enhance the manuscript's quality. A number of issues with language and structure are still of concern to the reviewers. Please take into consideration the points raised by the reviewers and make changes where recommended. If you wish to rebut a comment made by the reviewers, please explain this choice in your response letter with your resubmission. Reviewer #1 makes several, numbered points that require consideration. In your edits and responses to these points, I would particularly like a detailed justification in your letter for choices in response to points 1, 4, and 9 as these are of notable consequence to the manuscript's intellectual content and go far beyond language and style.

Reviewer 1 ·

Basic reporting

The revised manuscript is improved on the first version but there are still many problems with it and it is not suitable (yet) for publication. I note the problems below in order (roughly) of priority.

1. As requested, a detailed plan for eradication is now included but it omits improving quarantine regulations. There is no point in eradicating the fly if fresh root vegetables are still being imported. There will just be continuous invasions. Anyone who has the privilege of visiting the Antarctic must expect to be deprived of certain comforts that they enjoy in Europe. If these expeditioners are not prepared to live without fresh fruits and vegetables, then they should not accept a visit to the South Shetlands or any other Antarctic region. The Polish authorities should not put the desires of Polish scientists above that of the health of the environment. They should follow the lead of other European and nations elsewhere and ban the entry of fresh food to this region. I would suggest this must be an obvious recommendation included in this paper.
2. Why have the authors ignored the directions of referees as to correcting formatting? Hyphens are still being used when n dashes are the correct form. Moreover, hyphens have even been used when an m dash should be. It is quite annoying for referees who have spent some hours reviewing a mss to have their corrections ignored.
3. Although the English style has improved, there are still numerous instances of poor writing. I have again had to make corrections (see attached pdf).
4. I do not believe it is true that this is the first time an alien species has been shown to be transported within the Antarctic region. There are wind trap records both set on land and also towed behind aeroplanes in the region, that are published. None is referenced here.
5. The authors have cut some text and inserted what was asked for but there is still much repetition and the whole paper is far too long. The authors seem to be suffering what I can only describe as “verbal diarrhoea”. I repeat that for a fairly minor mss of this type, repeating information which is readily available on line or in other recent publications on a similar topic is not helpful for a reader not economical for the journal. There is also considerable repetition even within the text itself.
6. I have not checked the references and will not do so until the formatting is corrected.
7. Figure 4 is unnecessary and the morphological variation could be kept until the sequence data is available.
8. The authorities for species have still not been included the first time it is mentioned.
9. More discussion on why the attempts to eradicate the fly at Casey Station failed could be included i.e. what are the risks of the same failure on King George Island?
10. The authors do not say if there are any alien plants on the island. If so they could be a refuge for alien insects. Just a statement saying there are none would be sufficient.

Experimental design

n.a.

Validity of the findings

Adequate

Additional comments

See above

Annotated reviews are not available for download in order to protect the identity of reviewers who chose to remain anonymous.

·

Basic reporting

My concern here with the revision remains as previously that in some parts the language is still not up to the formal standard required by the journal, although it is much improved on the initial submission - even a cursory read of the abstract identifies several syntax errors that will require editorial correction, although this seems to be less of a problem with the rest of the ms.

Experimental design

As before, this is an observational study so experimental design is not relevant. The revised ms has been extensively altered from the original submission, in particular with several runs of text being deleted, especially following the suggestions of the other reviewer. This has improved the structure and flow of the ms.

Validity of the findings

Again, as before, this is an observational study of a novel and important observation, in the context of Antarctic biological invasions, and threats to native biodiversity. The revised ms is more focused and has improved logical flow.

Additional comments

a few remaining detailed comments:
l64 - spelling of Fildes Peninsula
l87/8 - were any found in these other locations?
l118 - it would be interesting to know the most northern 'natural' occurences of the species, as well as those that are synanthropic (and hence I assume in some sense 'introduced')?
l156 - I agree this knowledge is important, but perhaps even before that is the knowledge of where the species occurs - it is one thing to eradicate a species from within an enclosed sewage tank, but if it occurs naturally in the outside habitat the challenge becomes much greater if not impossible.
l161 - perhaps worth noting this is a very typical temperature range within the entire maritime Antarctic region - without pushing for self citation, see the detailed microclimate data presented and discussed in Convey et al 2018, Polar Biol. from several sites across the region, and also the simple distribution modelling for the also invasive species Eretmoptera murphyi presented by Hughes et al 2013 Biol Conserv.
l185 - is the species recorded from Spitsbergen as well as Bear Island? The former is not mentioned in earlier distributional text.
l187- worth noting that Kerguelen is a sub-Antarctic island which, as with all sub-Antarctic islands, faces much reduced seasonal thermal variation, and a much milder winter than more southern locations in the maritime Antarctic.
l193 - spelling of steinenii
l213/4 - the airfield also serves locations in the northern and southern Antarctic Peninsula, not only the South Shetland Islands.
l216/7 - there are wider references to the risks of transfer with cargo that would be appropriate to cite, in particular papers by JE Lee, and M Tsujimoto.
l245 - although the species hasn't been found in the sewage system of Escudero Station (which is a different station to Frei), I have personally caught adults inside windows in the station in January 2015 (if quoted this would be pers comm P Convey & T Contador) (same comment applies at l254)
l251 - 'identified' or 'confirmed' would be better than 'disclosed'
l279 - generally the IAATO biosecurity measures apply to visitor landings (mainly boot washing and kit cleaning), as these vessels do not normally discharge cargo. However, some cruise ships do send 'gifts' of fresh food to the stations they visit, so there may be a risk there.
l318 - the 'end of the 2017/18 season' has now passed, so has this eradication now been attempted, and what are the initial outcomes thought to be?
l354 - this is overstating what is said in the paper - as above, if the species is established in the natural environment, then none of the mentioned protocols will be appropriate to apply, and it is doubtfull whether any form of 'total eradication' is possible. This should be stated.

---

## Round 0.3 · accepted · Accept

At this time, I feel that your responses to reviewer comments are sufficient and you have adequately addressed their concerns. Efforts to improve the grammatical structure have enhanced the manuscript. Please be aware, however, that PeerJ does not offer copy editing services so it is important that you check your forthcoming proof for any overlooked typos or errors. Congratulations on this work.

#